# Genome-Wide Identification and Analysis on YUCCA Gene Family in *Isatis indigotica* Fort. and *IiYUCCA6-1* Functional Exploration

**DOI:** 10.3390/ijms21062188

**Published:** 2020-03-22

**Authors:** Miaomiao Qin, Jing Wang, Tianyi Zhang, Xiangyang Hu, Rui Liu, Tian’e Gao, Shuaijing Zhao, Yilin Yuan, Jinyu Zheng, Zirong Wang, Xiying Wei, Tao Li

**Affiliations:** National Engineering Laboratory for Resources Development of Endangered Crude Drugs in Northwest China, The Key Laboratory of Medicinal Resources and Natural Pharmaceutical Chemistry, The Ministry of Education, College of Life Sciences, Shaanxi Normal University, Xi’an 710119, China; qmm2018@snnu.edu.cn (M.Q.); wjwjgood123@gmail.com (J.W.); tianyizhang11@fudan.edu.cn (T.Z.); hu-xiangyang@snnu.edu.cn (X.H.); liurui_@snnu.edu.cn (R.L.); gtiane@snnu.edu.cn (T.G.); jing347431@snnu.edu.cn (S.Z.); yuanyilin@snnu.edu.cn (Y.Y.); zjingyu@snnu.edu.cn (J.Z.); wangzr@snnu.edu.cn (Z.W.); xiyingwei@snnu.edu.cn (X.W.)

**Keywords:** *Isatis indigotica* Fort., YUCCA, auxin, heterologous expression, stress tolerance

## Abstract

Auxin is one of the most critical hormones in plants. YUCCA (Tryptophan aminotransferase of *Arabidopsis* (TAA)/YUCCA) enzymes catalyze the key rate-limiting step of the tryptophan-dependent auxin biosynthesis pathway, from IPA (Indole-3-pyruvateacid) to IAA (Indole-3-acetic acid). Here, 13 YUCCA family genes were identified from *Isatis indigotica*, which were divided into four categories, distributing randomly on chromosomes (2*n* = 14). The typical and conservative motifs, including the flavin adenine dinucleotide (FAD)-binding motif and flavin-containing monooxygenases (FMO)-identifying sequence, existed in the gene structures. *IiYUCCA* genes were expressed differently in different organs (roots, stems, leaves, buds, flowers, and siliques) and developmental periods (7, 21, 60, and 150 days after germination). Taking *IiYUCCA6-1* as an example, the *YUCCA* genes functions were discussed. The results showed that *IiYUCCA6-1* was sensitive to PEG (polyethylene glycol), cold, wounding, and NaCl treatments. The over-expressed tobacco plants exhibited high auxin performances, and some early auxin response genes *(NbIAA8, NbIAA16, NbGH3.1*, and *NbGH3.6*) were upregulated with increased IAA content. In the dark, the contents of total chlorophyll and hydrogen peroxide in the transgenic lines were significantly lower than in the control group, with *NbSAG12* downregulated and some delayed leaf senescence characteristics, which delayed the senescence process to a certain extent. The findings provide comprehensive insight into the phylogenetic relationships, chromosomal distributions, and expression patterns and functions of the YUCCA gene family in *I. indigotica*.

## 1. Introduction

IAA (Indole-3-acetic acid) is the main type of natural auxin in plants, which plays important roles in a series of growth and developmental processes, such as embryonic polarization, vascular bundle differentiation, apical dominance, shoot architecture, root formation, phototropic response, gravity response, and senescence [1,2,3,4,5,6]. A growing body of evidence indicates that auxin participates in a variety of stress responses in plant, including high salt, low temperature, wounding, drought, and the like. Indeed, auxin can induce the rapid and transient high expression of auxin response genes *ARF, Aux/IAA, GH3, SAUR*, and *LBD*, which are involved in plant response to stress [7,8]. For example, *OsIAA24* and *OsIAA20* were up-regulated by high salt stress in *Oryza sativa* [9], and the expression level of *GH3.13* increased under drought treatment, which resulted in IAA content decreasing, with enhanced resistance to drought [10]. Furthermore, the sorting and intracellular cycling of *PIN2* were affected at a low temperature in *Arabidopsis thaliana*, which inhibited auxin transport in roots [11].

IAA is generally synthesized using tryptophan as a substrate through the IPA (indole-3-pyruvate synthesis pathway), IAM (indole-3-acetamide), IAOx (indole-3-acetaldoxime), and tryptamine pathways, of which the IPA synthesis pathway is the most important one [12]. The IPA pathway is divided into two steps; tryptophan is transformed into IPA by tryptophan aminotransferase (TAA), and then flavin monooxygenase (YUCCA), as the rate-limiting enzyme, catalyzes IPA to IAA [13,14,15]. Therefore, *YUCCA* genes play an important role in IAA biosynthesis process.

There were several research works reported on IAA accumulation and the regulation process. So far, 11 *YUCCA* genes were identified from *A. thaliana*, providing some evidence for the significant roles of *AtYUCCA1*, *AtYUCCA2*, *AtYUCCA4*, *AtYUCCA5*, *AtYUCCA6*, and *AtYUCCA7* in the biosynthesis of auxin [13,16,17,18,19]. Nevertheless, the function of YUCCA family genes is redundant. Over-expression of any one of *YUCCA1*, *YUCCA2*, *YUCCA5*, and *YUCCA6* in *A. thaliana* could result in a high-auxin phenotype; on the contrary, interference of single genes did not produce an auxin-deficient phenotype [13]. The *YUCCA6* activation mutant, *yuc6-1D*, presented an elevated IAA level and typical high-auxin phenotype with epinastic cotyledons, long hypocotyls, long narrow leaves, elongated petioles, and strong apical dominance [20]. It was not hard to find that the loss of function of multiple *YUCCA* genes caused developmental defects, while the over-expression of any one of *YUCCA* genes could promote auxin biosynthesis. When *YUCCA6* was over-expressed in *A. thaliana*, dark-induced senescence was dramatically delayed. Consequently, it was confirmed that auxin delayed senescence by directly or indirectly regulating the expression of senescence-associated genes (*SAG12*, *NAC1*, and *NAC6*) [16]. Over-expression of *AtYUCCA6* could increase the tolerance to 5 μM methyl viologen-mediated oxidative stress in *Ipomoea batatas* (L.) Lam [21], and the over-expressed auxin synthesis gene *YUCCA1* apparently delayed the senescence of strawberry fruit (*Fragaria* × *ananassa* Duch.) [22]. As a result, YUCCA family genes played profound roles in resisting stress in plants, and more research on the YUCCA gene family in more species was performed. Twelve *YUCCA* genes from *Populus trichocarpa* [23], 22 from *Glycine max* L. [24], 10 from *Citrullus lanatus* [25], 14 from *O. sativa* [26], 14 from *Zea mays* [27], three from *Triticum aestivum* [28], 10 from *Cucumis sativus* [29], 18 from *Brassica rapa* [30], and eight from *Fragaria vesca* [31] were identified or isolated, and their differential expression patterns were also characterized. The previous studies showed that YUCCA gene members shared highly conserved motifs, namely, the FAD-binding motif (GxGPxGLA), the nicotinamide adenine dinucleotide phosphate (NADPH)-binding motif (GxGxxGME), GC motif (ERxxxxASL), and ATG-containing motif (DxxxxATG) [24], which might be the pivotal sites for the biological functions of *YUCCA* genes. Furthermore, some evidence suggested that the FAD- and NADPH-binding motif GxGxxG is central to YUCCA activity [32].

*Isatis indigotica* Fort. is a biennial herb, just like *A. thaliana*, belonging to Cruciferae. Its roots and leaves have great medicinal and economic value. Clinically, its secondary metabolites have significant effects on influenza, epidemic encephalitis, hepatitis, mumps, and other diseases [33]. However, the auxin functions in regulating growth and development processes for *I. indigotica* are not as clear as for *A. thaliana* and other plants [12,16,17,20,24,25,26,27,28]. Due to the species differences in *YUCCA* genes, the phylogenetic, gene expression, and functional characterization of *IiYUCCA* genes could be similar or different, which needs to be further explored. Furthermore, it is extremely necessary to understand the molecular mechanism of auxin synthesis and regulation in *I. indigotica*. Based on the previous references of *YUCCA* genes in other species and the whole *I. indigotica* genome data (our unpublished data, another *I. indigotica* genome [34]), we collected the basic information of YUCCA gene family members and explored their expression patterns in different organs, different developmental periods, and different stress treatments. Moreover, the functions of *YUCCA* genes were discussed, taking *IiYUCCA6-1* as an example. The results provide comprehensive analyses on IiYUCCA gene family, with a firm foundation for further researches on their functions, adding to the data on *YUCCA* genes of Cruciferae.

## 2. Results

### 2.1. IiYUCCA Gene Family Analysis in I. indigotica

13 *YUCCA* genes in total were identified in *I. indigotica* (Table 1) and the *YUCCA* genes were named from *IiYUCCA1* to *IiYUCCA11* (*IiYUCCA1*, *IiYUCCA2*, *IiYUCCA3*, *IiYUCCA4*, *IiYUCCA5-1*, *IiYUCCA5-2*, *IiYUCCA6-1*, *IiYUCCA6-2*, *IiYUCCA7*, *IiYUCCA8*, *IiYUCCA9*, *IiYUCCA10*, *IiYUCCA11*), referring to the homologous YUCCA sequences in *A. thaliana*. Among them, *IiYUCCA6-2* was identified as a pseudogene, because there was a 2-bp deletion at the 450-bp position, which caused its translation to be terminated prematurely at the position of the 456-bp site when compared to *IiYUCCA6-1*. The CDS (Coding sequence) lengths of *IiYUCCAs* ranged from 1044 bp (*IiYUCCA10*) to 1335 bp (*IiYUCCA6-1*). Most *IiYUCCA* genes included three exons, while three genes (*IiYUCCA2*, *IiYUCCA3*, and *IiYUCCA6-1*) had four exons, and two genes (*IiYUCCA1* and *IiYUCCA4*) had five exons. The theoretical isoelectric point (pI) and molecular weight (Mw) ranged from 8.36 to 9.58 (average pI = 9.03) and from 38.64 kDa to 49.77 kDa (average Mw = 46.28 kDa), respectively. The detailed information of each *IiYUCCA* gene is listed in Table 1, including the gene identifier (ID) in the genome database, the length of the amino-acid sequence, chromosomal location, and some basic physical and chemical properties.

Based on the genome data, the genetic mapping of *IiYUCCA* genes on the chromosomes (2*n* = 14) was investigated (Figure 1). Except for *IiYUCCA6-1* and *IiYUCCA6-2* (which were not mapped to any chromosomes), the other 11 *IiYUCCA* genes were randomly distributed on five chromosomes: Chr01, Chr02, Chr03, Chr06, and Chr07. The distributions of *IiYUCCA* genes on the different chromosomes were not uniform. Chr06 had four *IiYUCCA* genes, the highest number, while Chr02 and Chr07 both had only one. Furthermore, there were three *IiYUCCA* genes on Chr03. It should be noted that *IiYUCCA5-1* and *IiYUCCA5-2* (located on Chr01) and *IiYUCCA3* and *IiYUCCA9* (located on Chr06) were very close to each other, and whether they might have similar biologic functions or not still needs to be confirmed.

### 2.2. Classification and Structural Analysis of IiYUCCA Genes

To clarify the structural diversities of *IiYUCCA* genes, an exon–intron composition map was drawn as shown in in Figure 2A. As expected, the gene family members in one branch exhibited similar intron–exon compositions, numbers, and lengths. For example, both *IiYUCCA10* and *IiYUCCA11* had three exons and similar coding sequence lengths, which indicated that they might have a close evolutionary relationship. Otherwise, 12 *IiYUCCA* genes (except for *IiYUCCA6-2*) were subjected to analysis with the MEME program to identify the conserved protein motifs. A total of 11 distinct motifs were found; among them, six motifs were identified. These were the FAD-binding motif, GC motif, ATG-containing motif 1, ATG-containing motif 2, FMO-identifying sequence, and NADPH-binding motif. It was evident that the closely related proteins shared the same motif profiles, which suggested that the IiYUCCA proteins within one branch might share conservative functions. Moreover, it was not difficult to find that the conserved motifs were nearly the same, both in terms of number and type. However, there were some differences between *IiYUCCA10* and *IiYUCCA11*. *IiYUCCA10* lacked ATG-containing motif 2 and motif 8 (Figure 2B). To some extent, the specific motifs might contribute to the functional divergences.

### 2.3. Phylogenetic Analysis of YUCCA Gene Family

In order to uncover the evolutionary relationships of YUCCA proteins among different species, we constructed the phylogenetic tree implemented in MEGA7.0, including *A. thaliana*, *Brassica rapa*, *Brassica oleracea*, *Raphanus sativus*, *O. sativa*, and *I. indigotica* (Figure 3). It was demonstrated that 96 YUCCA proteins from six species were grouped into four categories. Among them, Group I included five members, namely, YUCCA3, YUCCA5, YUCCA7, YUCCA8, and YUCCA9. The other groups contained two different members, such as YUCCA1 and YUCCA4 (Group II), YUCCA2 and YUCCA6 (Group III), and YUCCA10 and YUCCA 11 (Group IV). The phylogenetic analysis results apparently confirmed that YUCCAs of Cruciferae (dicotyledons) tended to be clustered into one branch, while *O. sativa* was separated from Cruciferae as monocotyledons. Therefore, it can be seen that the YUCCA proteins of *I. indigotica* were highly conserved evolutionarily. When the selection pressure was considered, the Ka/Ks ratio of the YUCCA gene family was 0.1126, far less than 1 (neutral selection) [33], which indicated a strong purifying selection on the YUCCA gene family for *I. indigotica*.

### 2.4. Expression Pattern Analysis on IiYUCCA Genes

The expression patterns of *IiYUCCAs* in different organs and different developmental periods (except *IiYUCCA5-2* and *IiYUCCA6-2*) were carried out. Here, *YUCCA5-2* and *YUCCA6-2* were more likely to be regarded as pseudogenes; thus, we did not determine their expression levels, but this does not mean that they have no biological function. As shown in Figure 4, all *IiYUCCA* gene expressions varied significantly in nine organs. It was notable that *IiYUCCA7* was specifically expressed in buds, which could play a key role in the bud development process. Interestingly, *IiYUCCA6-1* was expressed at relatively high levels in most organs, indicating its importance in *I. indigotica*. *IiYUCCA1*, *IiYUCCA9*, and *IiYUCCA11* were found to be predominantly expressed in the main roots, implying their specific functions in the root developmental process. *IiYUCCA4* and *IiYUCCA5-1* were mainly expressed in young leaves. *IiYUCCA2* and *IiYUCCA8* were expressed more highly in flowers than in other organs. Moreover, *IiYUCCA3* and *IiYUCCA10* were preferentially expressed in green siliques, reveling that they might participate in the reproductive development process. In summary, *IiYUCCA* members had various expression patterns in different organs, and they could play different roles in the accumulation and regulations of auxin in different organs of *I. indigotica*.

In addition to differences for different organs, the expression patterns in different developmental periods were also observed, and the four developmental periods (seven, 21, 60, and 150 days after germination) were detected. As shown in Figure 5, the expression levels of *IiYUCCA2*, *IiYUCCA4*, and *IiYUCCA10* increased gradually, whereas the levels of *IiYUCCA3*, *IiYUCCA5-1*, *IiYUCCA8*, and *IiYUCCA9* decreased with the developmental process. Moreover, *IiYUCCA11* showed a trend of decreasing firstly and then rising. *IiYUCCA1* expressions were detected only 21 days and 60 days after germination, whereas *IiYUCCA6-1* was expressed highly compared to other developmental periods. Therefore, these *YUCCA* genes had distinct spatiotemporal expression patterns in *I. indigotica*, laying a foundation for further function research on *IiYUCCA* genes.

Plants usually implement various signal regulatory networks to respond and adapt to different natural environments. Thus, it was important to understand what kinds of factors controlled the specific expressions of *YUCCA* genes and how they were regulated by different environmental signals. Due to the higher expression level in different organs, *IiYUCCA6-1* was taken as an example to examine expression patterns in response to various stress treatments. As shown in Figure 6, *IiYUCCA6-1* showed a strong response toward PEG treatment, and the expression changes firstly showed a sharp increase and then a rapid decrease. After 1 h of PEG treatment, the transcript level of *IiYUCCA6-1* was 1.13-fold higher than the control group. Additionally, the expression levels stimulated by NaCl showed no obvious changes, with a slight increase (0.21-fold) after 6 h of treatment. Furthermore, *IiYUCCA6-1* showed a completely opposite expression pattern when exposed to the wounding stimulation, and the expression level was 41% of the control group level after 1 h of treatment, before rising to 142%, demonstrating its sensibility to wounding stress. In addition, *IiYUCCA6-1* exhibited time-dependent and inconspicuous responses to the low-temperature treatment, but only demonstrated a 0.58-fold change after 12 h of treatment. In summary, *IiYUCCA6-1* exhibited the most obvious response to PEG treatment, but showed tiny variations when induced by low temperature. Therefore, the functions of the *IiYUCCA6-1* gene were correlated with the kinds of stress resistance to some extent.

### 2.5. Heterologous Expression of IiYUCCA6-1 in Nicotiana benthamiana

In this section, we further discuss *YUCCA* gene functions, taking *IiYUCCA6-1* as an example. *IiYUCCA6-1* was identified and cloned from *I. indigotica* (GenBank Accession Number: KY318462), encoding a protein highly homologous to *AtYUCCA6* (91%). The genome DNA of *IiYUCCA6-1* was 2835 bp, containing four exons and three introns. Its ORF (Open Reading Frame) length was 1335 bp, encoding 444 amino-acid residues. Detailed sequences and protein structure predictions are shown in the Appendix A. To determine the potential functions of *IiYUCCA6-1*, the gene was transformed into *N. benthamiana*. After screening with Hygromycin B, 10 transgenic lines (OE1–OE10) were tested at DNA and RNA levels. The detection results at the transcriptional levels demonstrated that *IiYUCCA6-1* was significantly expressed in 10 transgenic lines, and the expression levels of OE3 and OE9 were 200 times higher than in the control groups (Figure 7A). Free IAA content was also investigated in all over-expressed lines, and the contents increased significantly, except for OE1 and OE4 (Figure 7B), ranging from 5 to 6.5 μg/g. These results were not completely consistent with the expression changes of *IiYUCCA6-1*, indicating that *IiYUCCA6-1* may not be the only gene involved in the accumulation process of IAA.

Moreover, as one-month-old control groups, OE3 and OE9 plants were selected to observe the phenotypic differences. Obviously, the heights of OE3 and OE9 increased, and the whole plants were thin and weak with prominent apical dominance (Figure 7C), suggesting the typical hyperauxin phenotype. Meanwhile, the expression levels of four early auxin response genes, *NbIAA8*, *NbIAA16*, *NbGH3.1*, and *NbGH3.6*, were also observed, whose expressions were significantly higher than in the control groups (Figure 7D). These results illustrated that the early auxin response genes were upregulated, responding to the increased IAA content, which was upregulated by the over-expressed *IiYUCCA6-1* gene.

### 2.6. IiYUCCA6-1 Could Delay the Senescence Process

The fresh leaves of control and transgenic lines (OE3 and OE9) were isolated and treated with dark stress. The contents of chlorophyll and hydrogen peroxide and the expression of *NbSAG12* were detected. In Figure 8A, it is shown that the percentage of total chlorophyll content decreased significantly in the transgenic lines and reduced the leaf yellowing process after seven days of darkness treatment, which suggests that the senescence process of transgenic plants was apparently delayed. Moreover, the content changes of hydrogen peroxide were further detected. As displayed in Figure 8B, the increment in hydrogen peroxide content in transgenic lines was significantly lower than in the control group, which might have also contributed to the delay in the senescence process of transgenic plants. In addition, the expression level of *NbSAG12*, a senescence-related gene, was significantly lower than in the control group under darkness treatment (Figure 8C). These findings indicate that the senescence process was obviously reduced when *IiYUCCA6-1* was over-expressed in transgenic tobacco.

## 3. Discussion

### 3.1. YUCCA Numbers, Structures, and Phylogenetic Analysis in I. indigotica

As far as we know, most *YUCCA* genes in plants are gene families with multiple members, except for *Petunia*, which only has one *YUCCA* gene [35]. The YUCCA gene family was systematically described in *A. thaliana*, rice, maize, tomato, and so on [26,36,37,38,39,40]. Our study established some descriptions for the comprehensive features of the YUCCA gene family in *I. indigotica* for the first time. There are 13 *YUCCA* genes in *I. indigotica*, which are distributed on five chromosomes, but *IiYUCCA6-1* and *IiYUCCA6-2* were not located. Although *IiYUCCA6-1* and *IiYUCCA6-2* are very similar, the latter is a pseudogene. Therefore, it was speculated that *IiYUCCA6-1* might have vital functions because of the functional reduction of *IiYUCCA6-2*. Moreover, due to the similarities in both gene structures and conservative motifs, the *IiYUCCA* genes were deservedly clustered in one branch, showing the differences between *I. indigotica* and other plants (*A. thaliana*, *B. rapa*, *B. oleracea*, *R. sativus*, and *O. sativa*) based on the phylogenetic tree analysis. Furthermore, the *IiYUCCA* genes were divided into four subfamilies according to the conservative motifs. Meanwhile, the results showed that the YUCCA family members of several dicotyledons of Cruciferae were closely related and highly conserved in evolution, far from monocotyledon rice, consistent with the clustering analysis results of *OsYUCCA* gene family members in rice [41].

### 3.2. YUCCA Expression Patterns in I. indigotica

Gene expression patterns are usually consistent with their functions. Here, qRT-PCR analysis was conducted to investigate *IiYUCCA* gene expressions in nine organs and four developmental periods, which was conducive to assessing their possible biological functions. However, *IiYUCCA5-2* was not quantified by qRT-PCR, because its prediction exon sequence showed a base deletion at the 960-bp position, suggesting an early translation termination, hinting that *IiYUCCA5-2* might be a pseudogene. In addition, our transcriptome dataset of *I. indigotica* under different stress conditions (unpublished) showed no information for *IiYUCCA5-2* (Appendix A shows its specific nucleotide sequence). Overall, there were distinct expression differences between the vegetative (*IiYUCCA1*, *IiYUCCA4*, *IiYUCCA5-1*) and the reproductive organs (*IiYUCCA2*, *IiYUCCA7*, *IiYUCCA10*). Furthermore, *IiYUCCA9* and *IiYUCCA11* showed similar expression patterns, implying that there could be a certain degree of functional redundancy, which also contributed to stabilizing the genetic networks [24]. It was reported that the expression profiles of *YUCCA* genes in *P. trichocarpa* [23], *Lycopersicon esculentum* [40], and *Zea mays* [27] displayed a wide scope of expression patterns in different kinds of tissues. However, the *YUCCA* genes of *A. thaliana* [13] and *Fragaria vesca* [42] were mainly expressed in the meristem, young primordium, vascular tissue, and reproductive organs. Thus, the different expression profiles reflected the species differences, and the differences in *I. indigotica* need to be discussed. Some *IiYUCCA* gene expressions were upregulated along with the developmental process, such as *IiYUCCA2*, *IiYUCCA4*, *IiYUCCA10*, and *IiYUCCA11*. In contrast, the expression levels of *IiYUCCA2*, *IiYUCCA3*, *IiYUCCA5*, *IiYUCCA8*, and *IiYUCCA9* were downregulated. The results implied that these genes could play important roles in the developmental process. Therefore, the analyses of expression patterns in *IiYUCCA* genes provided some information to explore the functions of *YUCCA* genes in different organs and during different developmental processes, as well as contribute to speculating how spatio-temporal patterns of auxin biosynthesis are regulated in *I. indigotica*.

There was some evidence exhibiting that *YUCCA* genes performed potential regulatory functions under different kinds of stress conditions [17,43]. *IiYUCCA6-1* was sensitive to NaCl, PEG, cold, and wounding treatments. Its changes in expression pattern reflected that this gene might be involved in complex regulatory processes, with a function in defense responses. Similar results were found in *G. max*, whereas *C. sativus CsYUCCA8* and *CsYUCCA9* responded significantly to high temperature, and *CsYUCCA10b* was sensitive to low temperature and salt stress [29]. *GmYUCCA5* and *GmYUCCA8* were proven to be sensitive to different stress stimuli, including drought, copper, zinc, low temperature, high temperature, and salinity [24]. These findings indicated the potential functions of *YUCCA* genes in improving the stress-resistance capacities in plants. It is necessary to confirm the expression patterns of *YUCCA* genes in *I. indigotica*, which contribute to enriching the YUCCA gene family characteristics and functional researches in plants.

### 3.3. Functional Analysis on IiYUCCA6-1 Ectopically Expressed in Tobacco

Auxin plays important roles in cell division, lateral root formation, flowering time, and leaf senescence in plants [43]. The functions of the YUCCA gene family in IAA biosynthesis, plant development, and leaf senescence are typically associated with IAA level changes [16]. In this study, the functions of *IiYUCCA6-1* in the auxin biosynthesis process in *I. indigotica* were also discussed. Its functions in the IAA synthesis pathway were relatively consistent with those of *AtYUCCA6* reported in *A. thaliana* [22] and strawberry [42]. It was found that over-expression of *AtYUCCA6* increased not only IAA content, but also the expression levels of some auxin-responsive genes [22]. Here, similar results were obtained in transgenic tobacco, the expression levels of *NbIAA8*, *NbIAA16*, *NbGH3.1*, and *NbGH3.6* were significantly higher than those in the control group. The results demonstrated that the over-expressed *IiYUCCA6-1* gene might upregulate the auxin content, whereas the expression of early responsive genes was activated. Moreover, it was reported that YUCCA family genes could participate in plant resistance to stress [29], which was also proven in our experiment. Under dark stress, a visible change in total chlorophyll content in transgenic lines was observed, as well as H_2_O_2_ content. Furthermore, the expression level of *NbSAG12* decreased in the transgenic lines, which could be explained by the senescence process being delayed to a certain extent. Hence, the over-expression of *IiYUCCA6-1* in tobacco could promote IAA content, as well as upregulate the expression of auxin biosynthesis genes, which caused the typical hyperauxin phenotype. Furthermore, with a decrease in the contents of total chlorophyll and H_2_O_2_, as well as a downregulation of *NbSAG12*, the leaf senescence process was delayed, and stress resistance could also be improved. However, there are more stress conditions that still need further research.

## 4. Materials and Methods

### 4.1. Isolation and Sequence Analysis of YUCCA Gene Family in I. indigotica

To identify YUCCA family genes in *I. indigotica*, the nucleotide sequences of *YUCCA* genes from *A. thaliana*, *B. rapa*, *B. oleracea*, *R. sativus*, and *O. sativa* were downloaded from TAIR (https://www.arabidopsis.org/), brassicaDB (http://brassicadb.org), and RGAP (http://rice.plantbiology.msu.edu/). The BLASTn program (aligned a given nucleic acid sequence to the database) with a threshold of 1 × 10^−20^ was used to obtain the predicted sequences based on *I. indigotica* genome data. Then, Pfam (http://pfam.sanger.ac.uk/) and CDD (Conserved Domain Database) analyses were performed to confirm the encoded sequences of IiYUCCA proteins. GeneWise (Barcelona, Spain) was used to predict the pseudogenes. Furthermore, the physical and chemical properties (isoelectric point, molecular weight, GRAVY, and instability index) of IiYUCCA proteins were calculated using the ExPASy website (http://web.expasy.org/protparam/).

The chromosome locations of IiYUCCA members were investigated using MapChart software. The gene structures were analyzed using the Gene Structure Display Server (http://gsds.cbi.pku.edu.cn/, Peking University, Beijing, China). MEME software (http://meme-suite.org/, University of Nevada, Reno, USA) was used to find the common motifs, and the maximum number of motifs was set to 12 with the width of motifs set from six to 20. The multiple sequence alignments of YUCCA gene family proteins from six species were performed using the MUSCLE program (LynnonBiosoft Company, San Ramon, CA, USA) with default parameters. The full-length amino-acid sequences of 96 YUCCA proteins from six species were downloaded from their genome database, while the phylogenetic tree with aligned amino-acid sequences was constructed using MEGA 7.0 software with the maximum-likelihood method and verified with 1000 bootstraps. In addition, PAML 4.9d software was used to calculate the synonymous substitution rates (Ks) and non-synonymous substitution rates (Ka) of *YUCCA* genes in *I. indigotica*.

### 4.2. Plant Materials and Treatments

*I. indigotica* seeds were sown in pots (three plants each pot) in the greenhouse of the National Engineering Laboratory for Resources Development of Endangered Crude Drugs in Northwest China. The plants were cultivated at 23 ± 2 °C under a 16-h light/8-h dark photoperiod. Leaf materials at different developmental periods were taken from seedlings at seven days (cotyledon unfolding), 21 days (the second pair of euphylla unfolding), 60 days, and 150 days. Moreover, 60-day-old seedlings were treated with different abiotic stresses, namely, 5% PEG600, 100 mM NaCl, 4 °C, and 40% mechanical wounding, and the control group was irrigated with the same amount of deionized water. Then, all the leaf materials were collected after 0, 1, 3, 6, 12, and 24 h of treatment. The different organs, including roots, stems, leaves, buds, flowers, and siliques were taken from the plants in the field. All samples were quickly frozen in liquid nitrogen and stored at −80 °C for further research.

### 4.3. Gene Expression Analysis

Total RNA of each sample was extracted with TRIzol^®^ reagent (Invitrogen, Carlsbad, CA, USA) according to the manufacturer’s protocol. The quality of RNA was determined using a Nanodrop instrument (Thermo Fisher Scientific, Waltham, MA, USA) and agarose gel electrophoresis. The first-strand complementary DNA (cDNA) was synthesized using the PrimeScript TM II First Strand cDNA Synthesis Kit (TaKaRa, Dalian, China), with 1.0 µg of total RNA in 20 µL of water, stored at −20 °C. The RT-PCR products were diluted (1:20) for qRT-PCR analysis, which was performed using the SYBR^®^ Premix Ex Taq™ II (TaKaRa, Dalian, China) on a Roche LightCycler^®^96 System (Roche Diagnostics, Basel, Switzerland). All primer sequences for qRT-PCR are listed in the Appendix A. The endogenous reference genes from *I. indigotica*, *IieIF2*, *IiPP2A-4*, and *IiRPL15*, were used as the internal controls [44]. The relative gene expression levels were examined using the 2^−^^∆∆CT^ method. In each experiment, three biological and technical repeats were conducted.

### 4.4. Generation of Transgenic N. benthamiana

*IiYUCCA6-1* was cloned by the specific primers (forward (F): 5′–ATGGATTTCTGCT GGAGGA–3′, reverse (R): 5′–TCAACCGTCTAGTAATCTACTAT–3′), with the following PCR conditions: 94 °C, 3 min; 30 cycles of 94 °C, 30 s, 57 °C, 30 s, 72 °C, 2 min; 72 °C, 10 min. Subsequently, the product was cut with the restriction enzymes, *Bgl II* and *Spe I*, then inserted into the *pCAMBIA*1302 vector with the *CaMV35S* promoter. Furthermore, the vector was transformed into *Agrobacterium tumefaciens* strain *GV3101*, and the transformation of *N. benthamiana* was conducted using *Agrobacterium*-mediated leaf disc method. After screening by Hygromycin B, the over-expressed *IiYUCCA6-1* plants were sampled separately. One part of the plants was used to extract the genomic DNA, and the other part was used to extract RNA and reverse-transcribe to cDNA. Then, the transgenic lines were detected at the DNA and RNA level.

### 4.5. Heterologous Transformation of IiYUCCA6-1 to N. benthamiana

The fresh leaves from control and transgenic lines were selected for the gene function analysis. The cDNA was used to detect the expressions of early responsive genes (*NbIAA8, NbIAA16, NbGH3.1, NbGH3.6*), and free IAA contents were measured with the help of a Plant Indole-3-acetic acid ELISA Kit (Enzyme-linked Biotechnology Co., Ltd., Shanghai, China), following the manufacturer’s instructions. Next, some leaves were soaked in 3 mM MES (2-(N-morpholine) ethanesulfonic acid) medium (pH 5.7), and then placed in the dark. The total contents of chlorophyll and H_2_O_2_ (Jiancheng, Nanjing, China), and the expression level of senescence related gene *NbSAG12* (F: 5′–ATGAAGATGTGCCAGCG AAC–3′; R: 5′–AGCTACTCCCGTCTATTGCC–3′) were detected after dark treatments (zero and seven days), with *NbActin* (F: 5′–CGTTATGGTTGGAATGGGACAGAA–3′; R: 5′–AAGAACAGGGTGCTCCTC GTGG–3′) as the reference gene.

### 4.6. Statistical Analysis

Statistical analysis was performed with SPSS 22.0 software (IBM Company, Foster City, CA, USA). All experimental data were assessed using one-way analysis of variance (ANOVA), and results were presented as the means ± standard error (SE) at the levels of * *p* < 0.05, ** *p* < 0.01, and *** *p* < 0.001.

## 5. Conclusions

In brief, this study systematically analyzed the YUCCA gene family structures, phylogenetic characteristics, and expression patterns in *I. indigotica*. One of YUCCA gene family members, *IiYUCCA6-1*, could promote IAA accumulation and the upregulation of some auxin-responsive genes, which triggered a typical hyperauxin performance. Moreover, the contents of total chlorophyll and H_2_O_2_ decreased and the expression of *NbSAG12* was downregulated, which could delay the senescence process while improving the stress resistance (Figure 9). This study provided a systematic and comprehensive analysis of the *I. indigotica* YUCCA gene family using phylogenetic analysis, gene expression analysis, and functional characterization by heterologous expression in tobacco. All the findings demonstrated the experimental basis for the identification of the YUCCA genes in *I. indigotica*, with an exploration of their function.

## Figures and Tables

**Figure 1 ijms-21-02188-f001:**
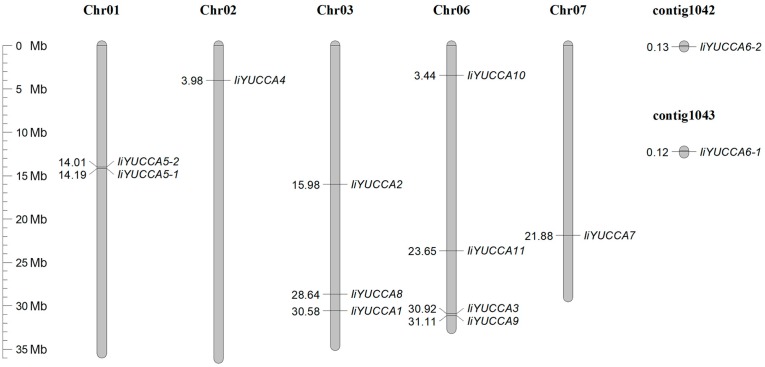
The chromosomal locations of *YUCCA* genes from *I. indigotica*. The scale represents a 35-Mb chromosomal distance. The numbers above represent the location of each gene.

**Figure 2 ijms-21-02188-f002:**
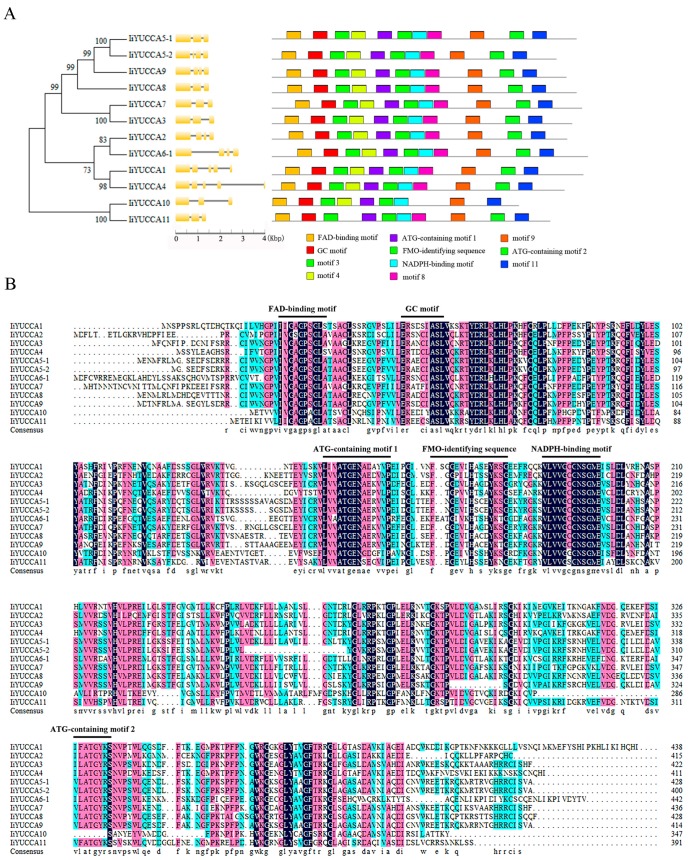
The phylogenetic tree, gene structure, multiple motifs (**A**), and multiple sequence alignment (**B**) of *YUCCA* gene family in *I. indigotica*. The gene structures of exons and introns are represented by yellow boxes and lines, respectively. The multiple conserved motifs were identified by MEME software, and they are labeled with colored boxes.

**Figure 3 ijms-21-02188-f003:**
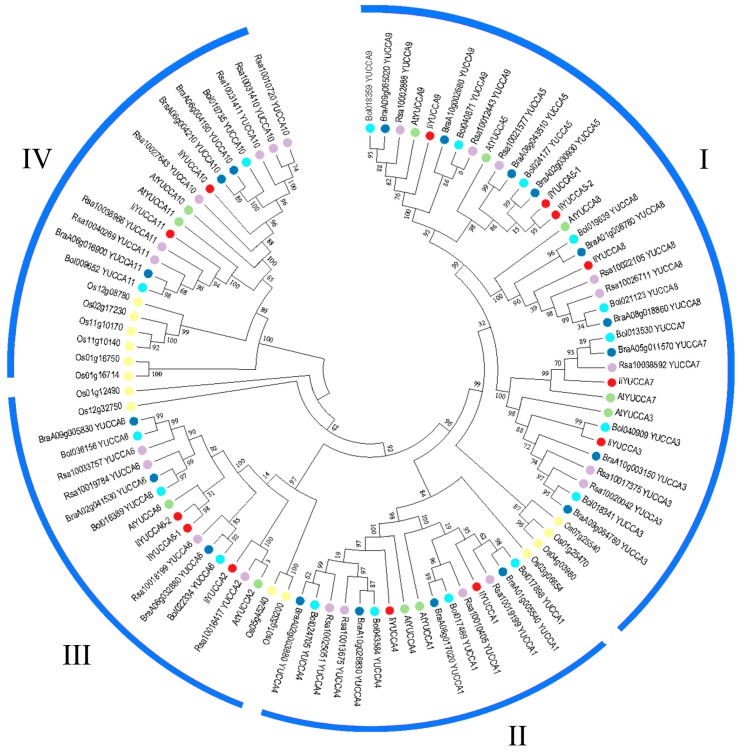
Phylogenetic relationships of *YUCCA* genes among *Arabisopsis thaliana* (green), *Brassica rapa* (dark blue), *B. oleracea* (light blue), *Raphanus sativus* (purple), *Oryza sativa* (yellow), and *I. indigotica* (red). The unrooted phylogenetic tree was constructed by MEGA 7.0.

**Figure 4 ijms-21-02188-f004:**
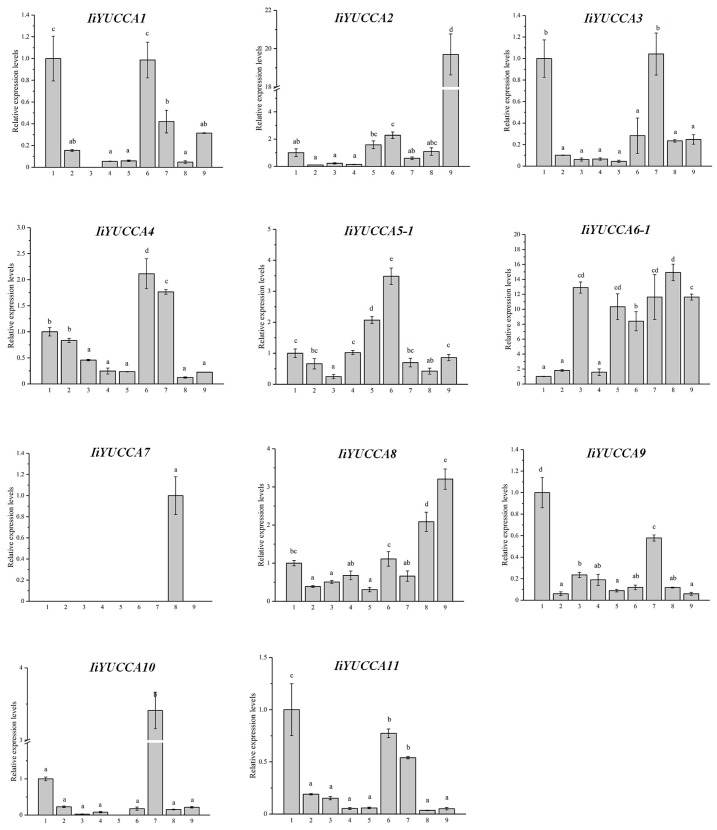
The expression patterns of *YUCCA* genes in different organs from *I. indigotica*. 1–9: mature root, lateral root, mature stem, young stem, mature leaf, young leaf, fruit, bud, and flower, respectively. Different lowercase letters indicate the significant differences for the relative expression levels at *p* < 0.05 level.

**Figure 5 ijms-21-02188-f005:**
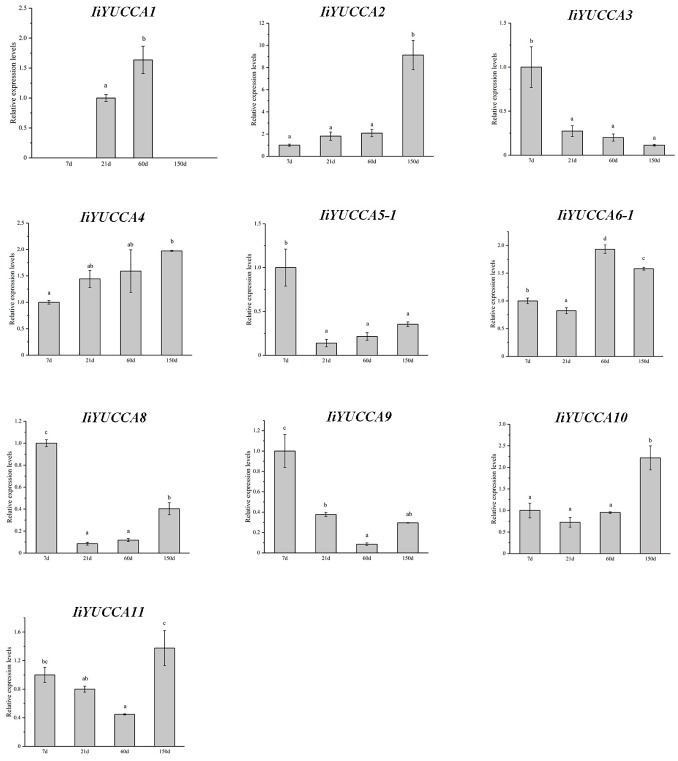
The expression patterns of *YUCCA* genes in different developmental periods from *I. indigotica*. Different lowercase letters indicate the significant differences for the relative expression at *p* < 0.05 level.

**Figure 6 ijms-21-02188-f006:**
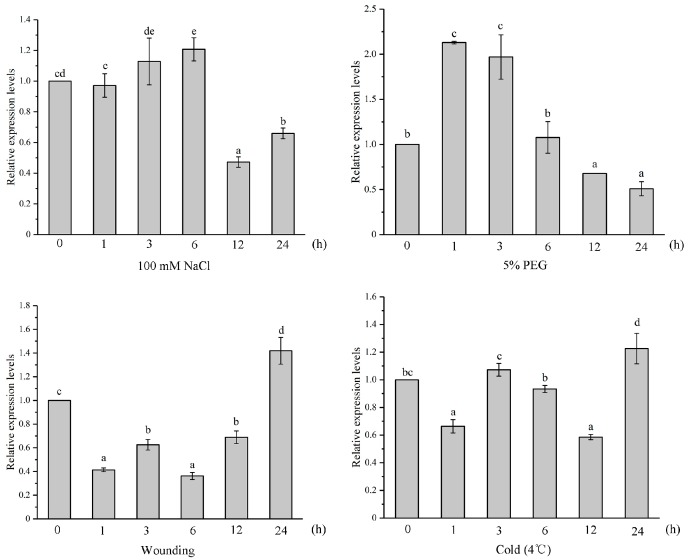
The expression patterns of *IiYUCCA6-1* under different stress treatments. Different lowercase letters indicate significant differences for the relative expression at the *p* < 0.05 level.

**Figure 7 ijms-21-02188-f007:**
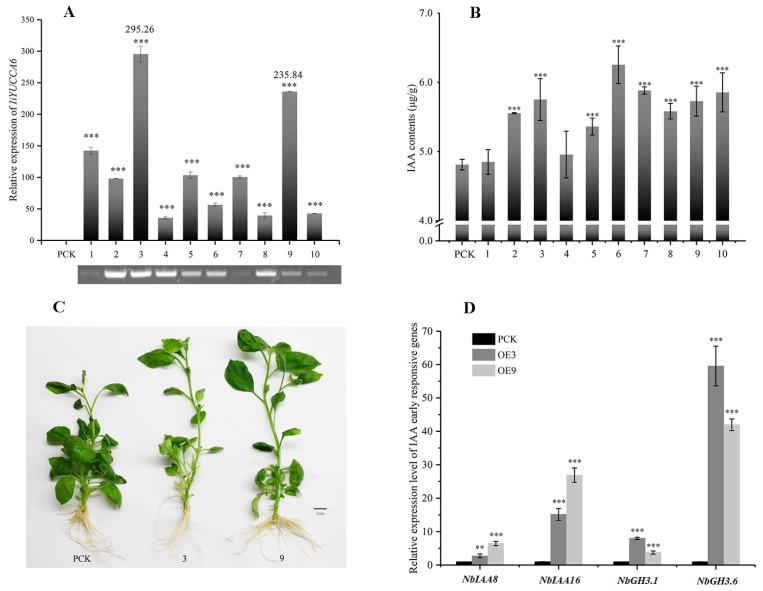
Identification and determination of the over-expressed *IiYUCCA6-1* in *N. benthamiana* (OE1–OE10). (**A**) DNA and RNA detections of transgenic *N. benthamiana*; (**B**) IAA content determinations in *N. benthamiana*; (**C**) the morphologic changes of transgenic *N. benthamiana*; (**D**) the expressions on IAA early responsive genes in transgenic lines. Significant differences are described as * *p* < 0.05, ** *p* < 0.01 and *** *p* < 0.001.

**Figure 8 ijms-21-02188-f008:**
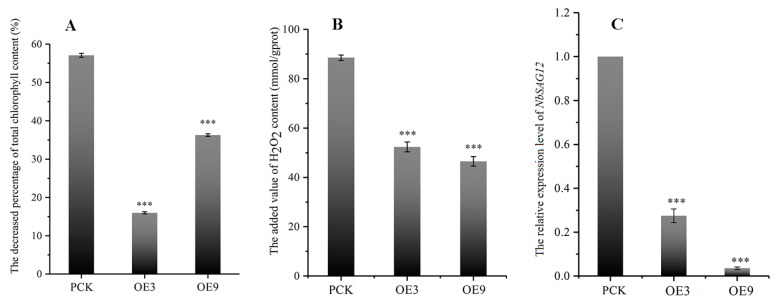
Determination of total chlorophyll and H_2_O_2_ contents and the expression of *NbSAG12*. (**A**) The decreased percentage of total chlorophyll content; (**B**) the increased values of H_2_O_2_ content; (**C**) the expression levels of *NbSAG12* in the transgenic lines.

**Figure 9 ijms-21-02188-f009:**
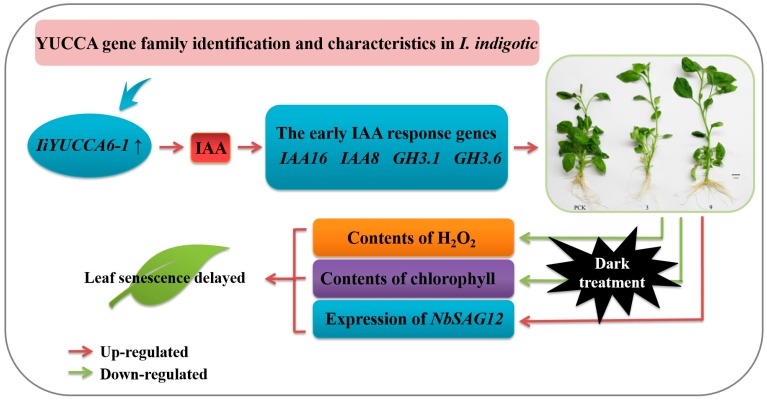
A model of *IiYUCCA6-1*-regulated IAA accumulation and leaf senescence.

**Table 1 ijms-21-02188-t001:** YUCCA family genes in *I. indigotica*.

Gene Name	Gene ID	Chromosome Location	CDS Length (bp)	Exons	Protein Length (aa)	M_W_ (Da)	pI	GRAVY	Instability Index
*IiYUCCA1*	EVM0033712	Chr06: 30576212–30578748	1317	5	438	48,658.38	9.58	−0.218	40.24
*IiYUCCA2*	EVM0000868	Chr03: 15976139–15977849	1248	4	415	46,528.12	8.36	−0.116	45.62
*IiYUCCA3*	EVM0030530	Chr06: 30914630–30916362	1269	4	422	47,162.38	9.15	−0.243	35.7
*IiYUCCA4*	EVM0029692	Chr02: 3979445–3983464	1236	5	411	45,732.06	9.39	−0.123	33.44
*IiYUCCA5-1*	EVM0003921	Chr01: 14186499–14187976	1287	3	428	47,701.11	9	−0.18	51.45
*IiYUCCA5-2*	EVM0010459	Chr01: 14011394–14012852	1203	3	400	44,461.14	8.91	−0.255	50.61
*IiYUCCA6-1*	EVM0022538	contig1043: 115626–118449	1335	4	444	49,773.53	8.71	−0.201	39.95
*IiYUCCA6-2*	Pseudogene492	contig1042: 131146–133992	456	-	-	-	-	-	-
*IiYUCCA7*	EVM0024488	Chr07: 21875662–21877324	1311	3	436	48,814.38	9.07	−0.207	42.26
*IiYUCCA8*	EVM0005748	Chr03: 28640430–28641928	1287	3	428	48,092.61	8.95	−0.225	48.86
*IiYUCCA9*	EVM0000072	Chr06: 31112478–31113966	1245	3	414	46,277.41	9.32	−0.243	47.12
*IiYUCCA10*	EVM0002314	Chr06: 3442231–3444777	1044	3	347	38,643.58	8.53	−0.122	26.13
*IiYUCCA11*	EVM0031632	Chr06: 23649486–23650852	1176	3	391	43,467.14	9.44	−0.154	42.9

Note: ID-identifier; aa-amino acid; M_W_-molecular weight; pI-isoelectric point. GRAVY-the grand average of hydropathy.

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
