# Peer review of "Genome-Wide Identification and Analysis on YUCCA Gene Family in Isatis indigotica Fort. and IiYUCCA6-1 Functional Exploration"

_ijms, 2020, doi:10.3390/ijms21062188_

Round 1

Reviewer 1 Report

The authors have provided a systematic and comprehensive analysis of I.indigotica YUCCA gene family by phylogenetic analysis, gene expression analysis and functional characterization by heterologous expression in Tobacco. I recommend this manuscript to be accepted for publication in International journal of molecular sciences.

Author Response

Dear Reviewer:

    Thank you for your comments on our manuscript entitled "Genome-wide identification and analysis on YUCCA gene family in Isatis indigotica Fort. and IiYUCCA6-1 functional exploration (IJMS-732878)". And we appreciate your evaluation and approval on our manuscript.

    We have already checked the language of this article carefully and corrected the spelling mistakes. The revised portions are marked with “Track Changes” in the manuscript. And the revisions will not influence the main contents and framework of the manuscript.

    Thank you for your comments and suggestions. We do appreciate for your warm work earnestly, and hope that the corrections will meet with approval.

    Best regards.

Response to Reviewer 1 Comments

Point 1:  English language and style are fine/minor spell check required.

Response 1: Thank you for your advice. As you mentioned, there were minor spelling mistakes in our manuscript. We have already checked the language of this article carefully and corrected the spelling mistakes. In addition, we invited professionals to check and correct errors in spelling, punctuation, and grammar for our manuscript again, and made the revision marked with “Track Changes” in it.

      We have tried our best to revise our manuscript according to your comments. And we’d like to submit for your kind consideration.

      Thanks a lot and best wishes.

Reviewer 2 Report

Review Comments

This study provided a comprehensive insight into the phylogenetic relationships, chromosomal distributions, and expression patterns of YUCCA gene family in I. indigotica. Furthermore, IiYUCCA6-1 played important roles in auxin biosynthesis and stress tolerance.

The study shows a sort of novelty. However, there is much information missed in this manuscript as shown below;

The authors should revise the abstract and add such important information to be understandable for the readers.

The introduction also missed the literature and up-to-date information covering this topic. The research question of the manuscript is not mentioned clearly. The authors should revise the introduction again and resolve this matter, including citing the literature in more details.

The manuscript still requires English language editing.

The discussion still needs improvement and should be interpreted to highlight the significant findings and how these findings will help to improve the knowledge on this topic.

-References

Recent literature should be cited to reveal the research significance and research question as well.

Author Response

Dear Reviewer:

    Thank you for your comments on our manuscript entitled "Genome-wide identification and analysis on YUCCA gene family in Isatis indigotica Fort. and IiYUCCA6-1 functional exploration (IJMS-732878)". The comments are valuable and very helpful to revise and improve our manuscript, with the important guiding significance to our further research works. We have studied all the comments carefully and made correction seriously, and we we hope the revisions could meet with your approvals. The revised portions are marked with “Track Changes” in the manuscript. And the revisions will not influence the main contents and framework of the manuscript.

    Thank you for your comments and suggestions. We do appreciate for your warm work earnestly, and hope that the corrections will meet with approval.

    Best regards.

Response to Reviewer 2 Comments

Point 1: The authors should revise the abstract and add such important information to be understandable for the readers.

Response 1: Thanks for your advice on the abstract and we totally agree your suggestion on this part. We have revised the abstract and add some important information to be understandable for the readers. The detailed modifications were described as:

Line 18-19: We supplemented the specific organs and named the developmental period, the revision was as follows: “different organs (roots, stems, leaves, buds, flowers and siliques) and developmental periods (7, 21, 60 and 150 days after germination)”

Line 21-26: We added the names of auxin response genes, and check the spelling and revised the languages. The revision was as follows: “The over-expressed tobacco plants exhibited high auxin performances, and some early auxin response genes (NbIAA8, NbIAA16, NbGH3.1 and NbGH3.6) were up-regulated, with increased IAA contents. In the dark, the contents of the total chlorophyll and hydrogen peroxide in the transgenic lines were significantly lower than the control group, and NbSAG12 down-regulated and some delaying leaf senescence characteristics, which delayed the senescence process to a certain extent”.

We have revise our manuscript according to your comments. We’d like to submit for your kind consideration and the manuscript could prompt readers to understand the paper better than before.

Point 2: The introduction also missed the literature and up-to-date information covering this topic.

Response 2: The introduction laid important foundations for understanding the research object of this paper. According to your comments, we added the up-to-date information covering the topic, and the revision was described as following, and marked it with “Track Changes” in the manuscript.

Line 33: “shoot architecture” was added to enrich IAA biological function statements.

Line 36-37: “Indeed, auxin can induce rapid and transient high expression of auxin response gene ARF, Aux/IAA, GH3, SAUR and LBD, which are involved in plant response to kinds of stress environments.” was added into this paragraph, according to the references.

Line 57-59: “It is not hard to find that the loss functions of multiple YUCCA genes could cause developmental defects, instead that over-expression of YUCCA genes could promote auxin biosynthesis.” was supplemented, which clearly explained the function of YUCCA gene in Arabidopsis.

Line 62-65: “Over-expression of AtYUCCA6 could increase the tolerance to 5 μM of methyl viologen-mediated oxidative stress in Ipomoea batatas (L.) Lam, and the over-expressed auxin synthesis gene YUCCA1 apparently delayed the senescence of strawberry fruit (Fragaria ×ananassa Duch.)” was added. In order to show the conservation and unity of the function of YUCCA genes, two examples that YUCCA gene participated in plant anti-stress processes in recent years were supplemented.

Line 69: “18 from Brassica rapa, and 8 from Fragaria vesca” was supplemented. With the development of gene sequencing technology, genome data of many species have been published. The numbers of YUCCA gene of two common species were added here, especially Brassica rapa, belonging to the same family as I. indigotica.

Line 73-74: “And some evidences suggested that the FAD- and NADPH-binding motif GxGxxG is central to YUCCA activity” was added. The latest evidence showed that the key domain of YUCCA genes was related to the conservation of its function, which could provide key information for further researches.

Point 3: The research question of the manuscript is not mentioned clearly.

Response 3: In order to provide the research questions of the manuscript clearly, we revised the introduction seriously again. We presented the direct and latest background references on YUCCA gene family research works in Cruciferae and other plants, and explore the function research of YUCCA genes. Due to the species differences of YUCCA genes, the phylogenetic, gene expression and functional characterization of IiYUCCA genes could be similar or different, which need to be further discussed. And the detailed results and analysis were presented in this paper.

Point 4: The manuscript still requires English language editing.

Response 4: As you mentioned, the manuscript still required English language editing. Thanks for your comments. To be honest, there were some spelling and grammar mistakes in this manuscript. We have combed the language carefully and corrected the spelling mistakes. In addition, we also invited professionals to check the grammatical errors for us. And we have made the revision marked with “Track Changes” in paper.

Point 5: The discussion still needs improvement and should be interpreted to highlight the significant findings and how these findings will help to improve the knowledge on this topic.

Response 5: Thanks for your advices on the discussion revision. We also agree that the discussion is the most important part of the paper, which made deep exploration for the research questions of the paper. Based on your comments, we made corresponding revisions for the discussions, which also marked with “Track Changes” in paper. Here, some repeat, improper or wrong expressions were deleted and modified, some new references were replenished to enrich the research works, and some statements were supplemented to highlight the research significances, and so on.

Certainly, there are still existed some spaces in this manuscript writings and research methods and results, we will try our best to study more from you, the other reviewer, editor and other researchers. There are many scientific questions worthy to be explored in this field, and we will peruse the works continually.

        Thank you and your comments, and we’d like to appreciate the opportunity to communicate our research works with others.

Point 6: Recent literature should be cited to reveal the research significance and research question as well.

Response 6: Thanks for your suggestion on the citation of recent literatures, which contributed to reveal the research significance and research question as well. The recent literatures cited were exhibited as follows:

[5], [6], [7] were used to supplement auxin functions in plant growth and anti-stress process.

[20], [21] were added to clarify the senescence-delayed function of YUCCA genes in other plants, such as delaying the ripeness of strawberry fruit.

[29], [30] were used to supplement the identification of YUCCA gene family in other species.

[31] was added to update the latest research on functional domain of YUCCA genes.

[34] was the latest published genome information of I. indigotica, which was released in February, 2020.

Point 7: The methods described inadequately.

Response 7: Thanks for your comments on the methods of this manuscript. The detailed modifications were as follows:

Line 339: The specific items of physical and chemical properties (isoelectric point, molecular weight, GRAVY and instability index) were added to make readers obtain YUCCA information easily. 

Line 346-348: We explained the source of 96 amino acid sequences, and described the steps of building phylogenetic tree in detail, so that the readers had an overall impression on the phylogenetic tree.

      Line 366-367: We supplemented the RNA test to make sure the RNA quality was good for further research.

Line 383-386: We have supplemented the screening and identification methods for the transgenic tobacco plants, which correspond to the results in Figure 7.

Line 389-390: In order to match the results in Figure 8, we supplemented the detection method for the transgenic plants, including the expression levels of early auxin responsive genes (NbIAA8, NbIAA16, NbGH3.1, NbGH3.6).

Point 8: The conclusions didn’t support by the results.

Response 8:  Thanks for your comments on the conclusions. To be honest, there is some inconsistency between the conclusion and the results, so we made some revisions for the conclusion, and the obtained results were supplemented as Line 405-408. In addition, the research significance in Line 408-410 was also provided. All the revisions contributed to the integrity and promotion of this paper.

     The detailed revisions were as follows:

Line 405-408: “some auxin-responsive genes up-regulated, which triggered typical hyperauxin performance. Moreover, the contents of total chlorophyll and H2O2 decreased and NbSAG12 down-regulated, which could delayed the senescence process and the stress resistance also improved” was supplemented.

Line 408-410: “This study provided a systematic and comprehensive analysis of I. indigotica YUCCA gene family by phylogenetic analysis, gene expression analysis and functional characterization by heterologous expression in tobacco.” was supplemented.

Round 2

Reviewer 2 Report

The review comments have been taken into consideration by the authors.